

# Linking ladder operators for the Rosen-Morse and Pöschl-Teller systems

**Simon Garneau-Desroches[1*] and Véronique Hussin[2,3]**

**1** Département de Physique, Université de Montréal, QC, H3C 3J7, Canada
**2** Département de Mathématiques et de Statistique,
Université de Montréal, QC, H3C 3J7, Canada
**3** Centre de Recherches Mathématiques, Université de Montréal, QC, H3C 3J7, Canada

* simon.garneau-desroches@umontreal.ca

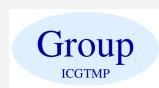

## Abstract

An analysis of the realizations of the ladder operators for the Rosen-Morse and Pöschl-Teller quantum systems is carried out. The failure of the *algebraic method* of construction in the general Rosen-Morse case is exposed and explained. We present the reduction of a recently obtained set of $(2n \pm 1)$-th-order Rosen-Morse ladder operators to the usual first-order realization for the Pöschl-Teller case known in the literature.

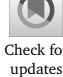
# 1 Introduction

Ladder operators are objects of fundamental importance in the context of exactly solvable quantum systems. While they connect adjacent eigenspaces of the Hamiltonian, they appear as key operators in the definition of coherent and squeezed states extensively studied for their properties in quantum optics, for instance [1]. Ladder operators also participate in describing the underlying structure of the system through the spectrum generating algebra (SGA) [2]. For the most common one-dimensional (1D) exactly solvable systems (harmonic oscillator, infinite square well, Morse, etc.), a systematic method has been developed to obtain a realization of the ladder operators as first-order differential operators [2]. However, more elaborated one-dimensional exactly solvable systems like the Rosen-Morse system fall outside the range of application of this algebraic method. This system, originally introduced as a model to study vibrations of polyatomic molecules, has been studied in different contexts recently [3–6]. Indeed, two different ladder operators realizations have been proposed in the literature, both as higher-order differential operators [4,5]. The first of which was motivated by an analogy with classical mechanics [6]; while the second arises purely from quantum mechanics through the concept of shape invariance in supersymmetric quantum mechanics (SUSYQM) [7].

In this paper, we investigate why realization as first-order differential operators of Rosen-Morse ladder operators cannot be achieved with the standard algebraic method. We find that the rational dependence of the bounded eigenstates parameters on the excitation number is crucial in providing an explanation. Moreover, the Rosen-Morse system is a generalization of the Pöschl-Teller system for which first-order ladder operators are known [8]. We then address the natural question of relating these sets of ladder operators. Starting from the most recent set of higher-order Rosen-Morse ladder operators, we explicitly show how they reduce to the known first-order realization in the Pöschl-Teller limit.

The present work is linked to the study of ladder operators in the context of SUSYQM and of exactly solvable systems of the Pöschl-Teller type on a larger scale. Indeed, ladder operators have been constructed and studied for rational extensions of the harmonic oscillator [9, 10] and of the Rosen-Morse I and II systems [4], among others. Besides, reflectionless cases of the Pöschl-Teller systems investigated in the following paper intervene in the context of soliton physics and have shown to exhibit non-linear supersymmetries [11, 12] similar to that of the rational extensions.

The plan of the paper is as follows. In Section 2, we review the Rosen-Morse and Pöschl-Teller exactly solvable systems. Then, we introduce ladder operators in Section 3 together with the algebraic method for obtaining realizations as first-order differential operators. The resulting ladder operators are presented for the Pöschl-Teller while the failure of the method is demonstrated for the general Rosen-Morse system. In Section 4, the construction of the most recent realization of higher-order ladder operators for the Rosen-Morse system is exposed. Then, we show explicitly in Section 5 how the higher-order ladder operators for the Rosen-Morse system reduce to their usual first-order realization in the Pöschl-Teller case. We make final conclusions in Section 6.

# 2 The Rosen-Morse and Pöschl-Teller systems

The (hyperbolic) Rosen-Morse (RM) [13] system is an exactly solvable quantum system with Hamiltonian labelled by the parameters $s$ and $\lambda$:

$$H_{s,\lambda} = -\frac{d^2}{dx^2} + 2\lambda \tanh(x) - s(s+1)\operatorname{sech}^2(x), \quad x \in \mathbb{R}, \quad s > 0, \quad 0 \le \lambda < s^2. \quad (1)$$

This system is also named Rosen-Morse II in the literature, as opposed to its trigonometric analogue (Rosen-Morse I) [14]. The particular case $\lambda = 0$ is the Pöschl-Teller (PT) system [2, 15]. The normalizable eigenstates solving the time-independent Schrödinger equation $H_{s,\lambda}\psi_{s,\lambda}(n) = E_{s,\lambda}(n)\psi_{s,\lambda}(n)$ are given in terms of the Jacobi polynomials $P_n^{(\alpha,\beta)}(y)$ [19]:

$$\psi_{s,\lambda}(n; x) = M_{s,\lambda}(n) \cosh^{-(s-n)}(x) e^{-\frac{\lambda x}{s-n}} P_n^{(a_{s,\lambda}(n), b_{s,\lambda}(n))}(\tanh(x)), \tag{2}$$

where the parameters are

$$a_{s,\lambda}(n) = s - n + \frac{\lambda}{s-n}, \qquad b_{s,\lambda}(n) = s - n - \frac{\lambda}{s-n}, \tag{3}$$

and $M_{s,\lambda}(n)$ is a normalization constant. There exists a finite number of bounded eigenstates and the associated energies are rational in the excitation:

$$E_{s,\lambda}(n) = -(s-n)^2 - \frac{\lambda^2}{(s-n)^2}, \qquad n = 0, 1, \ldots, n_{\max} < s - \sqrt{\lambda}. \tag{4}$$

The energies are related to the parameters through $E_{s,\lambda}(n) = -[a_{s,\lambda}^2(n) + b_{s,\lambda}^2(n)]/2$. In the Pöschl-Teller case, $a_{s,0}(n) = b_{s,0}(n)$ are linear in $n$ and the eigenstates can be expressed in terms of the associated Legendre polynomials $P_l^{\mu}(y)$ [19]. Moreover, the energy spectrum becomes quadratic in $n$.

## 3   Ladder operators and the algebraic method

In this work we define ladder operators $\{A^{\pm}(n)\}_{n=0}^{n_{\max}}$ for a given Hamiltonian $H$ by the following action on the bounded eigenstates:

$$A^{\pm}(n)\psi(n; x) \propto \psi(n \pm 1; x), \qquad A^-(0)\psi(0; x) = 0. \tag{5}$$

Here, $n_{\max}$ is either finite or infinite depending on $H$. They connect eigenspaces of adjacent energies: $A^+(n)$ is referred to as a raising operator and $A^-(n)$ is a lowering operator. This definition allows for different realizations of the ladder operators for a unique given system. Indeed, the proportionality constant can be chosen arbitrarily either to close an algebra or to construct certain types of coherent states, for instance. In the Rosen-Morse case, the bounded spectrum is finite and the action $A^+(n_{\max})\psi(n_{\max}; x)$ yields an unbounded state; we refer to [4] for more details.

For numerous exactly solvable systems (harmonic oscillator, infinitely deep square-well, Morse potential, etc.), there exists a standard technique to realize the ladder operators as first-order differential operators using the action on the eigenstates. This technique is sometimes referred to as the algebraic method in the literature [16, 17]. In particular, it has shown to be efficient for the Pöschl-Teller system. Starting with the assumption that $A^{\pm}(n)$ may be realized as

$$A^{\pm}(n) = g^{\pm}(n; x) + f^{\pm}(n; x)\frac{\mathrm{d}}{\mathrm{d}x}, \tag{6}$$

we act on an eigenstate $\psi(n; x)$ in order to get (5). The result is well-known and detailed in this case (see [2], for example). Indeed, ladder operators are found to be given as

$$A_{PT}^{\pm}(n) \propto -(s-n)\sinh(x) \pm \cosh(x)\frac{\mathrm{d}}{\mathrm{d}x}. \tag{7}$$

Let us now try to apply this technique to the Rosen-Morse case. We will show that it fails to obtain (5) in a straight way.

The idea is first to act with a derivative on the eigenstate, and then to use functional relations among the eigenstates to express the result in terms of the adjacent eigenstates. We take the usual change of variable $z = \tanh(x)$ and act with $\frac{d}{dx} = \text{sech}^2(x)\frac{d}{dz}$ on an eigenstate of (2). This yields

$$
\begin{aligned}
\frac{d}{dx}\psi_{s,\lambda}(n;x) = {} & \left[-(s-n)\frac{\sinh x}{\cosh x} - \frac{\lambda}{s-n}\right]\psi_{s,\lambda}(n;x) \\
& + M_{s,\lambda}(n)\cosh^{-(s-n)}(x)e^{-\frac{\lambda x}{s-n}}\text{sech}^2(x)\frac{d}{dz}P_n^{(a_{s,\lambda}(n),b_{s,\lambda}(n))}(z).
\end{aligned}
\tag{8}
$$

Now, one wants to make use of the functional relation [18]

$$
\frac{d}{dz}P_n^{(a_{s,\lambda}(n),b_{s,\lambda}(n))}(z) = \frac{2s-n+1}{2}\,P_{n-1}^{(a_{s,\lambda}(n)+1,b_{s,\lambda}(n)+1)}(z),
\tag{9}
$$

to recover $P_{n-1}^{(a_{s,\lambda}(n-1),b_{s,\lambda}(n-1))}(z)$ in order to have $\psi_{s,\lambda}(n-1;x)$ in (8). However, recalling the expression (3) for $a_{s,\lambda}(n)$ and $b_{s,\lambda}(n)$, this is only possible in the Pöschl-Teller case:

$$
\left.\begin{aligned}
a_{s,\lambda}(n)+1 &= a_{s,\lambda}(n-1) \\
b_{s,\lambda}(n)+1 &= b_{s,\lambda}(n-1)
\end{aligned}\right\} \qquad \Longleftrightarrow \qquad \lambda = 0.
\tag{10}
$$

Therefore, one cannot recover the eigenstate $\psi_{s,\lambda}(n-1;x)$ using this relation in the general Rosen-Morse setting. In fact, the problem comes directly from the rational dependence of the parameters $a_{s,\lambda}(n)$ and $b_{s,\lambda}(n)$ with respect to the excitation number $n$. Indeed, the functional relations that share the Jacobi polynomials $P_n^{(\alpha,\beta)}(z)$ only allow integer shifts of the parameters $\alpha$ and $\beta$ [19]. The same problem occurs when trying to recover $\psi_{s,\lambda}(n+1;x)$ instead. Consequently, the algebraic method fails to provide ladder operators for the Rosen-Morse system. The next section summarizes the most recent alternative way of constructing ladder operators for the Rosen-Morse system [4].

## 4 Ladder operators for the Rosen-Morse system

To simplify notation, we omit the explicit $x$-dependence of the eigenstates and use $\psi_{s,\lambda}(n)$ from this point on. We apply first-order supersymmetric (SUSY) transformation (see [7, 20], for example) to the Rosen-Morse Hamiltonian $H_{s,\lambda}$ and the corresponding eigenstates $\psi_{s,\lambda}(n)$. We get the so-called intertwining first-order differential operators

$$
B_{s,\lambda}^{\pm} = -s\tanh(x) - \frac{\lambda}{s} \pm \frac{d}{dx}.
\tag{11}
$$

The Rosen-Morse system $H_{s,\lambda}$ is known to be shape invariant with SUSY partner $H_{s-1,\lambda}$ with translated parameter $s \to s-1$ [7]. We have the usual eigenstates connections

$$
\psi_{s-1,\lambda}(n) = \frac{B_{s,\lambda}^{-}\psi_{s,\lambda}(n+1)}{\sqrt{E_{s,\lambda}(n+1)-E_{s,\lambda}(0)}}, \qquad \psi_{s,\lambda}(n+1) = \frac{B_{s,\lambda}^{+}\psi_{s-1,\lambda}(n)}{\sqrt{E_{s,\lambda}(n+1)-E_{s,\lambda}(0)}},
\tag{12}
$$

together with the ground state annihilation $B_{s,\lambda}^{-}\psi_{s,\lambda}(0) = 0$. The energies are preserved under the application of $B_{s,\lambda}^{\pm}$ as $E_{s-1,\lambda}(n) = E_{s,\lambda}(n+1)$. Successive applications of the SUSY transformation generate a hierarchy of Rosen-Morse Hamiltonians with fixed $\lambda$ and translating $s$. Since the system loses its ground state energy at every step of the procedure, the state $\psi_{s,\lambda}(n)$ of the initial system is connected to the ground state of the system $H_{s-n,\lambda}$ and vice versa:

$$
\psi_{s-n,\lambda}(0) \propto \left(B_{s-n+1,\lambda}^{-}B_{s-n+2,\lambda}^{-}\cdots B_{s-1,\lambda}^{-}B_{s,\lambda}^{-}\right)\psi_{s,\lambda}(n),
\tag{13}
$$

$$
\psi_{s,\lambda}(n) \propto \left(B_{s,\lambda}^{+}B_{s-1,\lambda}^{+}\cdots B_{s-n+2,\lambda}^{+}B_{s-n+1,\lambda}^{+}\right)\psi_{s-n,\lambda}(0).
\tag{14}
$$

$$H_{s,\lambda} \qquad H_{s-1,\lambda} \qquad \cdots \qquad H_{s-n+1,\lambda} \qquad H_{s-n,\lambda} \qquad H_{s-n-1,\lambda}$$

$$\psi_{s,\lambda}(n+1) \xleftarrow{B^+_{s,\lambda}} \psi_{s-1,\lambda}(n) \xleftarrow{B^+_{s-1,\lambda}} \cdots \xleftarrow{B^+_{s-n+2,\lambda}} \psi_{s-n+1,\lambda}(2) \xleftarrow{B^+_{s-n+1,\lambda}} \psi_{s-n,\lambda}(1) \xleftarrow{B^+_{s-n,\lambda}} \psi_{s-n-1,\lambda}(0)$$

$$\psi_{s,\lambda}(n) \xrightarrow{B^-_{s,\lambda}} \psi_{s-1,\lambda}(n-1) \xrightarrow{B^-_{s-1,\lambda}} \cdots \xrightarrow{B^-_{s-n+2,\lambda}} \psi_{s-n+1,\lambda}(1) \xrightarrow{B^-_{s-n+1,\lambda}} \psi_{s-n,\lambda}(0) \qquad \gamma_{s-n,\lambda}$$

$$\psi_{s,\lambda}(n-1) \xleftarrow{B^+_{s,\lambda}} \psi_{s-1,\lambda}(n-2) \xleftarrow{B^+_{s-1,\lambda}} \cdots \xleftarrow{B^+_{s-n+2,\lambda}} \psi_{s-n+1,\lambda}(0) \xleftarrow{\gamma^{-1}_{s-n+1,\lambda}}$$

Figure 1: Product decomposition of $A^\pm_{s,\lambda}(n)$ acting on $\psi_{s,\lambda}(n)$ to obtain $\psi_{s,\lambda}(n \pm 1)$ through the Rosen-Morse shape invariance hierarchy scheme.

Furthermore, defining

$$\gamma_{s,\lambda} = \cosh(x) e^{-\frac{\lambda x}{s(s-1)}}, \tag{15}$$

we obtain the ground states connections

$$\psi_{s-1,\lambda}(0) \propto \gamma_{s,\lambda} \psi_{s,\lambda}(0), \qquad \psi_{s+1,\lambda}(0) \propto \gamma^{-1}_{s+1,\lambda} \psi_{s,\lambda}(0), \tag{16}$$

which respectively raise and lower the value of the energy (4) in the hierarchy.

A ladder operator is constructed by applying successive intertwining operators from (13) on $\psi_{s,\lambda}(n)$ until a ground state is reached, then applying the connection (16), and finally applying successive intertwining operators from (14) to climb back in the hierarchy until $\psi_{s,\lambda}(n\pm 1)$ is reached [4]. The ladder operators $A^\pm_{s,\lambda}(n)$ write as the $(2n\pm 1)$-th-order differential operators

$$A^+_{s,\lambda}(n) \propto \left( B^+_{s,\lambda} B^+_{s-1,\lambda} \cdots B^+_{s-n+1,\lambda} B^+_{s-n,\lambda} \right) \gamma_{s-n,\lambda} \left( B^-_{s-n+1,\lambda} B^-_{s-n+2,\lambda} \cdots B^-_{s-1,\lambda} B^-_{s,\lambda} \right), \tag{17}$$

$$A^-_{s,\lambda}(n) \propto \left( B^+_{s,\lambda} B^+_{s-1,\lambda} \cdots B^+_{s-n+3,\lambda} B^+_{s-n+2,\lambda} \right) \gamma^{-1}_{s-n+1,\lambda} \left( B^-_{s-n+1,\lambda} B^-_{s-n+2,\lambda} \cdots B^-_{s-1,\lambda} B^-_{s,\lambda} \right). \tag{18}$$

The previous equations are valid with the exception of $A^-_{s,\lambda}(0)$, $A^+_{s,\lambda}(0)$ and $A^-_{s,\lambda}(1)$ for which they do not hold. For the latter two, one of the products should be interpreted as unity:

$$A^+_{s,\lambda}(0) \propto B^+_{s,\lambda} \gamma_{s,\lambda}, \qquad A^-_{s,\lambda}(1) \propto \gamma^{-1}_{s,\lambda} B^-_{s,\lambda}. \tag{19}$$

The particular case $A^-_{s,\lambda}(0)$ is also of the first order. For consistency with (18)[1] we take

$$A^-_{s,\lambda}(0) \propto \gamma^{-1}_{s+1,\lambda} B^-_{s,\lambda,}, \tag{20}$$

even though $B^-_{s,\lambda}$ already annihilates $\psi_{s,\lambda}(0)$. The ladder operators $A^\pm_{s,\lambda}(n)$ satisfy the action (5) and they are illustrated in Figure 1 where their action is decomposed within the hierarchy.

## 5 Reduction of Pöschl-Teller ladder operators

This section addresses the reduction of the Rosen-Morse ladder operators $A^\pm_{s,\lambda}(n)$ to the known Pöschl-Teller first-order realization $A^\pm_{PT}(n)$ presented in Section 3. We set $\lambda = 0$ and remove the $\lambda$-label so that $A^\pm_s(n), B^\pm_s, \gamma_s$ and $\psi_s(n)$ are understood to be that of the Pöschl-Teller system.

---

[1] As well as for technical reasons in view of Section 5.

## 5.1   Reduction of $A_{s,0}^+(n)$

We show how the action of $A_s^+(n)$ reduces to that of $A_{PT}^+(n)$. To do so, we use ideas from [8]. Raising the $\psi_s(n)$ state develops as

$$A_s^+(n)\psi_s(n) \propto B_s^+ B_{s-1}^+ \cdots B_{s-n+1}^+ \left[ B_{s-n}^+ \cosh(x) B_{s-n+1}^- B_{s-n+2}^- \cdots B_{s-1}^- B_s^- \psi_s(n) \right], \qquad (21)$$

where the factor in brackets yields the state $\psi_{s-n}(1)$ (see Figure 1). Knowing the expression for $\psi_{s-n}(1)$, we write it in terms of the ground state of the same system in the hierarchy:

$$\psi_{s-n}(1) \propto \sinh(x) \cosh^{-(s-n)}(x) \propto \sinh(x) \psi_{s-n}(0). \qquad (22)$$

Substituting back in (21), we arrive at

$$A_s^+(n)\psi_s(n) \propto B_s^+ B_{s-1}^+ \cdots B_{s-n+1}^+ \sinh(x)\psi_{s-n}(0). \qquad (23)$$

The $\sinh(x)$ must be commuted to the left of the product of intertwining operators in order to recover $\psi_s(n)$ on the right hand side via the relation (14). This can be done by first writing the product of intertwining operators in the form

$$B_s^+ B_{s-1}^+ \cdots B_{s-n+1}^+ = \cosh^{s+1}(x) \left( \mathrm{sech}(x)\frac{\mathrm{d}}{\mathrm{d}x} \right)^n \cosh^{-(s-n+1)}(x), \qquad (24)$$

and then by making use of the commutation relation

$$\left[ \left( \mathrm{sech}(x)\frac{\mathrm{d}}{\mathrm{d}x} \right)^n, \sinh(x) \right] = n \left( \mathrm{sech}(x)\frac{\mathrm{d}}{\mathrm{d}x} \right)^{n-1}, \qquad (25)$$

among differential operators [8]. We obtain

$$A_s^+(n)\psi_s(n) \propto \sinh(x) B_s^+ B_{s-1}^+ \cdots B_{s-n+1}^+ \psi_{s-n}(0) + n \cosh(x) B_{s-1}^+ \cdots B_{s-n+1}^+ \psi_{s-n}(0) \qquad (26)$$

$$\propto \sinh(x) B_s^+ \psi_{s-1}(n-1) + n \cosh(x)\psi_{s-1}(n-1), \qquad (27)$$

where we again used (14) in the last line. Then, from (12), we use respectively

$$B_s^+ \psi_{s-1}(n-1) = \sqrt{n(2s-n)}\psi_s(n), \quad \text{and} \quad \psi_{s-1}(n-1) = \frac{B_s^- \psi_s(n)}{\sqrt{n(2s-n)}}, \qquad (28)$$

on the first and second terms of (27) to recover the action of a first-order differential operator on $\psi_s(n)$:

$$A_s^+(n)\psi_s(n) \propto \left[ \sqrt{n(2s-n)}\sinh(x) + \frac{n\cosh(x)}{\sqrt{n(2s-n)}} B_s^- \right] \psi_s(n). \qquad (29)$$

Simplifying using the expression for $B_s^-$, the remaining operator is proportional to the usual raising operator for the Pöschl-Teller system:

$$A_s^+(n)\psi_s(n) \propto \left[ -(s-n)\sinh(x) + \cosh(x)\frac{\mathrm{d}}{\mathrm{d}x} \right] \psi_s(n) \propto A_{PT}^+(n)\psi_s(n). \qquad (30)$$

In the general Rosen-Morse case, the operators $B_{s,\lambda}^\pm$ contain a $\lambda/s$ term which complicates the generalization of the identity (24). Then, the association (22) contains two terms with exponentials. Put together, the commutation of the $B_{s,\lambda}^\pm$ cannot be performed similarly and prevents the reduction of the ladder operators.

## 5.2 Reduction of $A^-_{s,0}(n)$

The reduction of $A^-_s(n)$ is similar to that of $A^+_s(n)$. Developing the lowering of the $\psi_s(n)$ state in a similar fashion as done in (21) and (22) yields

$$A^-_s(n)\psi_s(n) \propto B^+_s B^+_{s-1} \cdots B^+_{s-n+2} \, \mathrm{sech}(x) B^-_{s-n+1} \psi_{s-n+1}(1) . \tag{31}$$

Note that we have made the association with $\psi_{s-n+1}(1)$ before reaching the step of ground state connexion (see Figure 1). To continue further, we make use of the equivalence of the following operators on $\psi_{s-n+1}(1)$:

$$\left. \begin{array}{c} \mathrm{sech}(x) B^-_{s-n+1,\lambda} \\ \cosh(x) B^-_{s-n,\lambda} \end{array} \right\} : \ \psi_{s-n+1,\lambda}(1) \mapsto \psi_{s-n+1,\lambda}(0) , \tag{32}$$

to write

$$A^-_s(n)\psi_s(n) \propto B^+_s B^+_{s-1} \cdots B^+_{s-n+2} \cosh(x) B^-_{s-n} \psi_{s-n+1}(1) . \tag{33}$$

Noticing $B^-_{s-n} = B^-_{s-n+1} + \tanh(x)$, we get

$$A^-(n)\psi_s(n) \propto B^+_s B^+_{s-1} \cdots B^+_{s-n+2} \big[\cosh(x) B^-_{s-n+1} + \sinh(x)\big] \psi_{s-n+1}(1) . \tag{34}$$

We use $B^+_s \cosh(x) = \cosh(x) B^+_{s-1}$ repeatedly on the first term of (34) to commute $\cosh(x)$ to the left. The second term is treated similarly as in the previous section by using the formula (24) and the commutation relation (25) on $\sinh(x)$, yielding two terms. Keeping track of the relative constants between the terms, we obtain

$$\begin{aligned} A^-_s(n)\psi_s(n) \propto {}& \cosh(x) B^+_{s-1} \cdots B^+_{s-n+2} B^+_{s-n+1} B^-_{s-n+1} \psi_{s-n+1}(1) \\ & + \sinh(x) B^+_s B^+_{s-1} \cdots B^+_{s-n+2} \psi_{s-n+1}(1) \\ & + (n-1)\cosh(x) B^+_{s-1} \cdots B^+_{s-n+2} \psi_{s-n+1}(1) . \end{aligned} \tag{35}$$

The product $B^+_{s-n+1} B^-_{s-n+1}$ factorizes $H_{s-n+1}$ in the first term and the three terms can then be combined. We act with the product $B^+_{s-1} \cdots B^+_{s-n+2}$ to get $\psi_{s-1}(n-1)$ (see Figure 1). We are left with

$$A^-_s(n)\psi_s(n) \propto \big[(2s-n)\cosh(x) + \sinh(x) B^+_s\big] \psi_{s-1}(n-1) . \tag{36}$$

We again make use of (28) and rearrange to recover the usual Pöschl-Teller lowering operator

$$A^-_s(n)\psi_s(n) \propto \left[-(s-n)\sinh(x) - \cosh(x)\frac{\mathrm{d}}{\mathrm{d}x}\right]\psi_s(n) \propto A^-_{PT}(n)\psi_s(n) . \tag{37}$$

## 6 Conclusion

In this paper, we have studied ladder operators for the Rosen-Morse system and the Pöschl-Teller particular case. We exposed how the algebraic method of constructing ladder operators fails for the general Rosen-Morse system, and we found that the rational dependence of the parameters $a_{s,\lambda}(n)$ and $b_{s,\lambda}(n)$ on $n$ is responsible for this failure. Next, we recalled the construction of a set of known $(2n \pm 1)$-th-order Rosen-Morse ladder operators. It was expected that the later should reconcile with the well-known first-order realization of the Pöschl-Teller ladder operators obtained from the algebraic method. Indeed, we have explicitly obtained the reduction of the Pöschl-Teller ladder operators from order $2n \pm 1$ to order 1. Besides, a point canonical transformation [21] has been used to map the ladder operators presented in Section 4 onto $(2n \pm 1)$-th-order analogous ladder operators for the trigonometric Rosen-Morse system [4]. We expect that similar results apply in the trigonometric case.

# Acknowledgements

**Funding information** V. Hussin acknowledges the support of research grant from NSERC of Canada.

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
