# Peer review of "Linking ladder operators for the Rosen-Morse and Pöschl-Teller systems"

_SciPost Physics Proceedings, doi:SciPost Phys. Proc. 14, 026 (2023)_

## Round 1 · Referee Report · Anonymous (Referee 1) · 2023-3-20

Strengths

The presented results are interesting not only from the point of view of the considered quantum systems, but also due to their close connection with completely integrable systems (KdV hierarchy) and solitons.

Weaknesses

Two important points indicated in the report, unfortunately, escaped the attention of the authors.

Report

In the work, the construction of ladder operators for the Rosen-Morse and Pöschl-Teller systems is carried out. The presented results are interesting not only from the point of view of these quantum systems, but also due to their close connection with completely integrable systems (KdV hierarchy) and solitons. Two important points, unfortunately, escaped the attention of the authors.

1) It is known that there are broad classes of exactly solvable quantum systems characterized by a discrete spectrum resembling the spectrum of finite-gap quantum systems (DOI: 10.1088/1751-8121/aa739b , 10.1103/PhysRevD.98.026017 , 10.1103/PhysRevD.106.089901 ). As a consequence, they are characterized by the presence of a triple of basic (primary) pairs of the lowering and raising ladder operators, which form complete sets of spectrum generating operators. Such systems belong to the family of systems mentioned in the Introduction, and therefore this important feature from the point of view of physics should be pointed out.

2) The origin of such operators in rational deformations of a quantum harmonic oscillator and conformal mechanics bears some resemblance to a peculiar property of reflectionless hyperbolic Pöschl-Teller systems characterized by the presence of non-trivial Lax-Novikov integrals, which are the Darboux-dressed momentum operator of a free quantum particle (see DOI: 10.1016/j.aop.2006.12.002 , 10.1103/PhysRevLett.101.030403 , 10.1016/j.aop.2009.01.009 , 10.1007/JHEP12(2017)061 ). It is this nontrivial differential operator of higher odd order that underlies the exotic non-linear supersymmetry associated with (multi-)soliton potentials and their reflectionless nature, detects all bound states of the corresponding quantum systems, and distinguishes doubly degenerate states in the continuous parts of their spectrum. A special case of such systems corresponds to the Pöschl-Teller systems considered in the article (s=1,2,…). I suggest adding an appropriate comment related to the indicated feature of the Pöschl-Teller systems.

After appropriately taking into account the remarks, the article can be recommended for publication.

Requested changes

1) It is known that there are broad classes of exactly solvable quantum systems characterized by a discrete spectrum resembling the spectrum of finite-gap quantum systems (DOI: 10.1088/1751-8121/aa739b , 10.1103/PhysRevD.98.026017 , 10.1103/PhysRevD.106.089901 ). As a consequence, they are characterized by the presence of a triple of basic (primary) pairs of the lowering and raising ladder operators, which form complete sets of spectrum generating operators. Such systems belong to the family of systems mentioned in the Introduction, and therefore this important feature from the point of view of physics should be pointed out.

2) The origin of such operators in rational deformations of a quantum harmonic oscillator and conformal mechanics bears some resemblance to a peculiar property of reflectionless hyperbolic Pöschl-Teller systems characterized by the presence of non-trivial Lax-Novikov integrals, which are the Darboux-dressed momentum operator of a free quantum particle (see DOI: 10.1016/j.aop.2006.12.002 , 10.1103/PhysRevLett.101.030403 , 10.1016/j.aop.2009.01.009 , 10.1007/JHEP12(2017)061 ). It is this nontrivial differential operator of higher odd order that underlies the exotic non-linear supersymmetry associated with (multi-)soliton potentials and their reflectionless nature, detects all bound states of the corresponding quantum systems, and distinguishes doubly degenerate states in the continuous parts of their spectrum. A special case of such systems corresponds to the Pöschl-Teller systems considered in the article (s=1,2,…). I suggest adding an appropriate comment related to the indicated feature of the Pöschl-Teller systems.

---

## Round 2 · Author Response

We wish to thank the referee(s) for reviewing our paper and for the useful comments provided in the initial decision report. The remarks brought up in the report have contributed to the completeness of the paper and have engendered discussions that will influence future works. It is after studying the proposed references that we address this reply to inform the referees on the modifications made in the paper.

Both points brought by the referee(s) have been implemented in the paper as a paragraph in the introduction. The reason for this choice is to inform the reader on topics closely related to that discussed the present work and to point toward appropriate references. It is precisely the third paragraph of Section 1 that contains the comments.

A comment on the constructions of ladder operators in the case of rationally extended solvable systems was formulated. We chose to explicitly mention the rational extension of the harmonic oscillator since it is a very well documented rational extension for which ladder operators have been studied at different orders of SUSY transformations. We referred to the references 10.1088/1751-8121/aa739b and 10.1103/PhysRevD.98.026017 suggested in the review. We also took the opportunity to mention the ladder operators study on the rational extensions of both the trigonometric and hyperbolic Rosen-Morse systems performed in one of our previous papers: 10.1088/1751-8121/ac2549 .

Then, one comment was made regarding the reflectionless cases of the Pöschl-Teller system studied in our paper. We mentioned the highlighted non-linear supersymmetry it has shown to exhibit and we emphasized the relevance of this system in the field of soliton physics. We joined to that comment two of the references pointed by the referee(s): 10.1016/j.aop.2006.12.002 and 10.1016/j.aop.2009.01.009 .

Best regards,
The authors

---

## Round 2 · List of Changes

• The points brought by the referee(s) have been implemented as the third paragraph of Section 1.

  • A comment on the constructions of ladder operators in the case of rationally extended solvable systems was formulated.

  • The well documented rational extension of the harmonic oscillator was explicitly mentioned together with the references 10.1088/1751-8121/aa739b and 10.1103/PhysRevD.98.026017

  • The rational ladder operators for the Rosen-Morse systems from a previous work from the authors were mentioned.

  • The non-linear supersymmetry exhibited by the reflectionless Pöschl-Teller systems was mention as the paper discusses both the Pöschl-Teller system and supersymmetric quantum mechanics. References 10.1016/j.aop.2006.12.002 and 10.1016/j.aop.2009.01.009 .

  • Importance of this system in the study of soliton physics was emphasized.

---

## Editorial Decision

published